# Monthly sulfadoxine/pyrimethamine-amodiaquine or dihydroartemisinin-piperaquine as malaria chemoprevention in young Kenyan children with sickle cell anemia: A randomized controlled trial

Steve M. Taylor[1,2,3]*, Sarah Korwa[4], Angie Wu[2], Cynthia L. Green[2,5], Betsy Freedman[1], Sheila Clapp[2], Joseph Kipkoech Kirui[4], Wendy P. O'Meara[1,3], Festus M. Njuguna[4,6]

1 Division of Infectious Diseases, Duke University School of Medicine, Durham, North Carolina, United States of America, 2 Duke Clinical Research Institute, Durham, North Carolina, United States of America, 3 Duke Global Health Institute, Duke University, Durham, North Carolina, United States of America, 4 Academic Model Providing Access to Healthcare (AMPATH), Eldoret, Kenya, 5 Department of Biostatistics and Bioinformatics, Duke University School of Medicine, Durham, North Carolina, United States of America, 6 Department of Child Health and Paediatrics, School of Medicine, College of Health Sciences, Moi University, Eldoret, Kenya

* steve.taylor@duke.edu

## Abstract

### Background

Children with sickle cell anemia (SCA) in areas of Africa with endemic malaria transmission are commonly prescribed malaria chemoprevention. Chemoprevention regimens vary between countries, and the comparative efficacy of prevention regimens is largely unknown.

### Methods and findings

We enrolled Kenyan children aged 1 to 10 years with homozygous hemoglobin S (HbSS) in a randomized, open-label trial conducted between January 23, 2018, and December 15, 2020, in Homa Bay, Kenya. Children were assigned 1:1:1 to daily Proguanil (the standard of care), monthly sulfadoxine/pyrimethamine-amodiaquine (SP-AQ), or monthly dihydroarte-misinin-piperaquine (DP) and followed monthly for 12 months. The primary outcome was the cumulative incidence of clinical malaria at 12 months, and the main secondary outcome was the cumulative incidence of painful events by self-report. Secondary outcomes included other parasitologic, hematologic, and general events. Negative binomial models were used to estimate incidence rate ratios (IRRs) per patient-year (PPY) at risk relative to Proguanil. The primary analytic population was the As-Treated population. A total of 246 children were randomized to daily Proguanil ($n = 81$), monthly SP-AQ ($n = 83$), or monthly DP ($n = 82$). Overall, 53.3% ($n = 131$) were boys and the mean age was 4.6 ± 2.5 years. The clinical malaria incidence was 0.04 episodes/PPY; relative to the daily Proguanil group, incidence rates were not significantly different in the monthly SP-AQ (IRR: 3.05, 95% confidence interval [CI]: 0.36 to 26.14; $p = 0.39$) and DP (IRR: 1.36, 95% CI: 0.21 to 8.85; $p = 0.90$) groups.

**Data Availability Statement:** Data cannot be shared publicly because the signed informed consent form placed restrictions on sharing broadly. For inquiries of data use that fall within the scope of the study and therefore the consent authorization, please contact the study team at epitomise-study@duke.edu.

**Funding:** Supported by the NHLBI (www.nhlbi.nih. gov) of the NIH (R01 HL134211 to SMT). The funders had no role in study design, data collection and analysis, decision to publish, or preparation of the manuscript. SP-AQ was supplied free of charge to the trial by Guilin Pharmaceuticals, which had no role in the design, conduct, analysis, reporting, or decision to report the results.

**Competing interests:** The authors have declared that no competing interests exist.

**Abbreviations:** ALT, alanine aminotransferase; AT, As-Treated; ATP, According-to-Protocol; CI, confidence interval; CIR, cumulative IR; DBS, dried blood spot; DP, dihydroartemisinin-piperaquine; DSMB, Data Safety and Monitoring Board; ECG, electrocardiogram; EPiTOMISE, Enhancing Preventive Therapy of Malaria in children with Sickle cell anemia in East Africa; G6PD, glucose-6-phosphate dehydrogenase; HbSS, homozygous hemoglobin S; HPLC, high-performance liquid chromatography; HR, hazard ratio; IR, incidence rate; IRR, incidence rate ratio; ITT, Intention-To-Treat; LAR, legally authorized representative; ms, millisecond; NB, negative binomial; NHLBI, National Heart Lung and Blood Institute; PPY, per patient-year; QTcF, Fridericia's corrected QT interval; RDT, rapid diagnostic test; SAE, severe adverse event; SCA, sickle cell anemia; sCr, serum creatinine; SD, standard deviation; SP-AQ, sulfadoxine/pyrimethamine-amodiaquine.

Among secondary outcomes, relative to the daily Proguanil group, the incidence of painful events was not significantly different in the monthly SP-AQ and DP groups, while monthly DP was associated with a reduced rate of dactylitis (IRR: 0.47; 95% CI: 0.23 to 0.96; $p = 0.038$). The incidence of *Plasmodium falciparum* infection relative to daily Proguanil was similar in the monthly SP-AQ group (IRR 0.46; 95% CI: 0.17 to 1.20; $p = 0.13$) but reduced with monthly DP (IRR 0.21; 95% CI: 0.08 to 0.56; $p = 0.002$). Serious adverse events were common and distributed between groups, although compared to daily Proguanil ($n = 2$), more children died receiving monthly SP-AQ ($n = 7$; hazard ratio [HR] 5.44; 95% CI: 0.92 to 32.11; $p = 0.064$) but not DP ($n = 1$; HR 0.61; 95% CI 0.04 to 9.22; $p = 0.89$), although differences did not reach statistical significance for either SP-AQ or DP. Study limitations include the unexpectedly limited transmission of *P. falciparum* in the study setting, the high use of hydroxyurea, and the enhanced supportive care for trial participants, which may limit generalizability to higher-transmission settings where routine sickle cell care is more limited.

## Conclusions

In this study with limited malaria transmission, malaria chemoprevention in Kenyan children with SCA with monthly SP-AQ or DP did not reduce clinical malaria, but DP was associated with reduced dactylitis and *P. falciparum* parasitization. Pragmatic studies of chemoprevention in higher malaria transmission settings are warranted.

## Trial registration

clinicaltrials.gov (NCT03178643).
   Pan-African Clinical Trials Registry: PACTR201707002371165.

## Author summary

### Why was this study done?

- Sickle cell anemia (SCA) is a very common condition among children born in malaria-endemic areas of sub-Saharan Africa, but their supportive care regimens are poorly tailored to African settings.

- Among the complications that children with SCA suffer are more severe outcomes owing to malaria, and, therefore, many African countries recommend various malaria preventive regimens for children with SCA.

- It is important to compare the efficacy and safety of these regimens in order to enhance the supportive care of African children with SCA.

### What did the researchers do and find?

- In this study, 3 malaria chemoprevention regimens were compared among children under 10 years old with SCA at a single site in Homa Bay, Kenya.

- Children were randomly assigned to take daily Proguanil (which is the standard of care in Kenya), a monthly combination of sulfadoxine-pyrimethamine with amodiaquine (SP-AQ), or a monthly combination of dihydroartemisinin-piperaquine (DP), and then followed monthly for 12 months.

- Cases of malaria were very low among all 3 groups, but the combination of DP reduced the risk of being infected by *Plasmodium falciparum* parasites and of dactylitis, which is a common complication of SCA.

- DP was not associated with a higher rate of serious adverse events, but SP-AQ was associated with an unexpectedly higher rate of deaths that did not achieve statistical significance.

**What do these findings mean?**

- Monthly DP may be an alternative to existing chemoprevention regimens for children with SCA owing to its safety, acceptability, and efficacy on hematologic events.

- SP-AQ-associated mortality among children with SCA was unexpected and though it did not achieve statistical significance merits further evaluation.

- Further studies are needed to compare chemoprevention regimens on parasitologic and hematologic outcomes in areas of high *P. falciparum* transmission and delivered through routine SCA providers.

## Introduction

Sickle cell anemia (SCA) afflicts over 300,000 newborns annually [1], and most of these children will be born in malaria-endemic regions of sub-Saharan Africa. In African settings, the mortality rate of SCA children is 20 times that of non-SCA children [2], and up to 90% of newborns with SCA die before the age of 5 years [3]. Broadly, this severity is the result of both hematologic and infectious complications, including severe anemia, vaso-occlusive (painful) crises, bacteremia and bacterial pneumonias, and malaria. Additionally, many children lack access to routine supportive care for SCA, including early diagnosis, specialized follow-up, and provision of hydroxyurea and other prophylactic medications. A pressing need exists to design and implement comprehensive SCA care that is tailored to African settings.

Malaria is more severe and deadly in children with SCA [4,5] and can precipitate vaso-occlusive complications [6,7]. Therefore, many country-level guidelines recommend routine malaria chemoprevention, including daily Proguanil in Kenya [8] and Nigeria [9] and monthly sulfadoxine-pyrimethamine (SP) in Uganda [10]. These policies are supported by meta-analyses that suggest benefit [11,12], although there is a paucity of high-quality, comparative effectiveness studies of malaria chemoprevention regimens in African children with SCA.

We tested the efficacy of 3 chemoprevention regimens in Kenyan children with SCA: daily Proguanil, monthly sulfadoxine/pyrimethamine plus amodiaquine (SP-AQ), and monthly dihydroartemisinin-piperaquine (DP). Daily Proguanil is the standard of care in Kenya; monthly SP-AQ is efficacious in the African Sahel as Seasonal Malaria Chemoprevention [13]; and DP is highly effective as therapy and, in research studies, as monthly prevention in

children [14] and pregnant women [15]. We compared these 2 alternate regimens to the standard of care on the efficacy to prevent clinical malaria and adverse hematologic outcomes among children aged 1 to 10 years with laboratory-confirmed SCA in Western Kenya.

## Methods

### Trial design and oversight

The Enhancing Preventive Therapy of Malaria in children with Sickle cell anemia in East Africa (EPiTOMISE) trial was an open-label, parallel assignment randomized trial. The trial protocol was approved by ethics committees of Moi University (0001907) and Duke University (Pro00077428), registered with clinicaltrials.gov (NCT03178643) and the Pan-African Clinical Trials Registry (PACTR201707002371165), and approved by the Kenyan Pharmacy and Poisons Board (ECCT/17/08/06). The Duke Clinical Research Institute oversaw the trial, with technical and infrastructural support from the Duke Global Health Institute and Moi University School of Medicine. Written informed consent in English, Kiswahili, or Dholuo was obtained from the parents or guardians of all children. Independent study monitoring was performed biannually; a medical safety monitor reviewed all severe adverse events (SAEs); and a Data Safety and Monitoring Board (DSMB) convened by the National Heart Lung and Blood Institute (NHLBI) provided biannual review of progress and safety. SP-AQ was supplied free of charge to the trial by Guilin Pharmaceuticals, which had no role in the design, conduct, analysis, reporting, or decision to report the results. This study is reported as per the Consolidated Standards of Reporting Trials (CONSORT) guideline (S1 Checklist).

### Trial setting and population

EPiTOMISE was conducted in Homa Bay, Kenya, a town of historically high malaria transmission; in 2012 to 2013, 46% of children with fever presenting to Homa Bay County Hospital had malaria [16]. Children between 12 months and 10 years of age with homozygous hemoglobin S (HbSS) disease confirmed by hemoglobin electrophoresis and who met additional eligibility criteria were enrolled from a routine SCA clinic at Homa Bay County Hospital and randomized in 1:1:1 ratio to receive either daily Proguanil, monthly SP-AQ, or monthly DP in an open-label fashion. Randomization was performed through the electronic data capture program using a block allocation with a block size of 9. No stratification criteria were used.

The inclusion criteria for enrollment were as follows: age greater than 12 months and less than 10 years at enrollment; current attendance at or willingness to attend the study clinic; residence in either Homa Bay County or the Rongo or Awendo subcounties of Migori County; confirmed hemoglobin genotype of HbSS by electrophoresis or high-performance liquid chromatography (HPLC); no immediate, apparent, or reported plans to relocate residence in the next 2 years; ability to take oral medication and be willing to adhere to the medication regimen or caregiver willingness to give the medical regimen as prescribed; and ability and willingness of parent or legally authorized representative (LAR) to give informed consent. From May 2019 were added the criteria at screening of hemoglobin concentration $\geq$6.5 g/dL and alanine aminotransferase (ALT) $\leq$50 U/L. The exclusion criteria were as follows: taking routine antimalarial prophylaxis for another indication (including co-trimoxazole for HIV infection); known allergy or sensitivity to sulfadoxine, pyrimethamine, amodiaquine, proguanil, dihydroartemisinin, piperaquine, artemether, lumefantrine, penicillin (if under 5 years old), or derivatives of these compounds; known chronic medical condition other than SCA (i.e., malignancy, HIV) requiring frequent medical attention; currently participating in another clinical research study, or having participated in one in the prior 30 days; living in the same household as a previously enrolled study participant; chronic use of medications known to prolong the QT

interval in children; Fridericia's corrected QT interval (QTcF) >450 milliseconds (ms); or receipt of a transfusion of red blood cells in the 120 days prior to screening.

## Trial interventions

Following randomization to chemoprevention regimen, children were dispensed standard doses of each study drug at monthly follow-up visits, which were self-administered by caregivers; dosing (see **S1 Text**) was weight-based for Proguanil and DP and age-based for SP-AQ. Both SP-AQ (SPAQ-CO) and DP (D-Artepp) were acquired from a manufacturer with WHO prequalification (Guilin), the former supplied without charge by the manufacturer; Proguanil was manufactured by Cosmos Ltd (Nairobi, Kenya). Per local guidelines, children under 5 years also received daily penicillin at standard age-based doses, and all children were enrolled in the Kenya National Hospital Insurance Fund. Adherence to administration of prevention regimen was assessed at each monthly follow-up visit by trial staff via structured queries.

## Trial assessments

Caregivers were encouraged to call trial staff and report for an acute visit if the child was unwell. At these and at routine monthly visits, children with fever >38˚C or a history of subjective or objective fever in the prior 24 hours were tested for malaria with a rapid diagnostic test (RDT) detecting the parasite antigens HRP2 and pLDH (SD Bioline Malaria Ag P.f/Pan COMBO, Alere). Children with positive RDT results were treated with a standard weight-based course of artemether-lumefantrine.

At each monthly visit, we performed a complete review of systems and physical exam, assessed concomitant medications, and collected an interval history, including interval tests or treatments for malaria with verification when possible by review of outside records, self-reported painful crises and dactylitis, and SAEs. Venous blood was collected every 3 months for complete blood count, serum creatinine (sCr), and ALT and every 6 months for hemoglobin electrophoresis; clinical laboratories met ISO 15189 requirements. From May 2019, hemoglobin concentration was also measured at each visit using a point-of-care test (HemoCue Hb 301). At each acute care and routine visit, capillary blood was collected as a dried blood spot (DBS) for post hoc molecular detection of *Plasmodium falciparum*. In children allocated to DP residing in Homa Bay town, we repeated an electrocardiogram (ECG) 4 to 6 hours following the third dose of each monthly DP course.

## Outcome measures

All children were followed for 12 months after randomization. The primary outcome was clinical malaria incidence, defined as subjective or objective fever with the presence of *P. falciparum* parasites measured by a malaria RDT or by severe malaria using a standard definition [17]. Secondary parasitologic outcomes were hospitalization for malaria, light microscopy–positive malaria, unconfirmed malaria, and asymptomatic parasitization. The main hematologic outcome was the incidence of painful events, defined as an episode of pain lasting 2 hours or more without an obvious cause, and additional secondary hematologic outcomes were dactylitis, defined as pain or tenderness with or without swelling of the hands or feet, transfusion of blood products, severe anemia (defined as hemoglobin concentration <5.5 g/dL), and acute chest syndrome. See **S1 Text** for full outcome definitions. Safety outcomes included SAEs, ALT elevation, anemia, and, in DP recipients, prolongation of the QT interval corrected for heart rate by Fridericia's method (QTcF). *P. falciparum* was detected in all DBS using a real-time PCR assay [18]. In selected participants, glucose-6-phosphate dehydrogenase (G6PD) was genotyped by PCR amplification and bidirectional Sanger sequencing across

nucleotides 202 and 376 that encode the most common A- deficiency variant in East Africa [19,20], and polymorphisms in CYP2C8*2 were similarly genotyped (see **S1 Text** for molecular methods) [21,22].

## Statistical analysis

To estimate the sample size, we assumed 3.7 episodes per patient-year (PPY) using the malaria rate in 2011 by children enrolled in the RTS, S/AS01 vaccine trial in Siaya, Kenya [23], which historically has had similar transmission to Homa Bay [24], and a similar expected distribution of episodes shifted by treatment effect factors. Using simulations, we found enrolling 65 patients per treatment arm would provide >90% power to detect a reduction of 40% in the DP arm, approximately 40% power to detect a reduction of 20% in the Proguanil arm, and >90% power to detect a reduction of 40% in either arm. Allowing for 20% loss to follow-up, a sample size of 246 (82 per group) provided at least 90% power, assuming that each comparison between experimental arm and control would be tested at the 0.0269 level (Dunnett's correction) in order to preserve a nominal alpha level of 0.05.

The analysis was initially planned using the Intention-To-Treat (ITT) population. This was changed to the As-Treated (AT) population during the finalization of the statistical analysis plan owing to the unanticipated crossover from October 2018 to May 2019 from the SP-AQ group to Proguanil; this crossover resulted from the temporary suspension of SP-AQ administration at the request of the DSMB to allow assessment of 4 deaths among SP-AQ recipients. Participants who were still in active follow-up assigned to the SP-AQ arm in May 2019 were crossed back over to SP-AQ at that time. Analyses using the According-to-Protocol (ATP) were also evaluated to compare patients treated as randomized (e.g., did not crossover).

Results are presented by treatment arm using the mean, standard deviation (SD), median, 25th and 75th percentiles (Q1, Q3), and range as appropriate for continuous variables, as well as the count and percentage for non-missing data for categorical variables.

Primary and secondary incidence rates (IRs), or the number of episodes PPY at risk, for each experimental arm were compared to the control using a generalized regression model with a negative binomial (NB) distribution to allow for interdependence between multiple episodes. This model structure allowed the use of the actual follow-up time as the offset variable and a robust sandwich variance estimate. Model distributions considered included the NB, Poisson and zero-inflated Poisson, and NB with the distribution chosen based on the residual plots. Model results are presented using the incidence rate ratio (IRR) with 95% confidence interval (CI). Prespecified subgroup analyses explored the impact of changes to protocol and access to care prompted by the COVID-19 pandemic and other covariates on clinical malaria and painful events by including an interaction term in the model; the main pandemic-related protocol change was that monthly follow-up visits alternated between in-person and telephone visits. Additional covariates at enrollment included age, hydroxyurea use, sex, hospitalization in prior 12 months, and hemoglobin concentration. Secondary dichotomous outcomes are presented as cumulative incidence (CI) rates per treatment arm and compared to the control using Fine and Gray's method to allow death to be a competing risk and censoring of patients lost to follow-up within the ITT and ATP populations. Comparisons are presented using hazard ratio (HR) with 95% CI. The AT population was not evaluated for comparing cumulative IRs due to multiple crossing of treatment arms with events possibly occurring in each crossed arm. **S1 Analysis** describes all comparisons of the monthly groups to the daily Proguanil group; analyses comparing monthly DP to monthly SP-AQ were added later.

Each primary and secondary outcome comparison used an α = 0.0269 per Dunnett's test for multiple comparisons to the same control group. Tables were generated from complete

case data only. No imputation of baseline or endpoint data was done. All analyses were conducted using SAS 9.4 (SAS Institute, Cary, NC, USA).

## Results

### Participants and follow-up

The trial was conducted between January 23, 2018, and December 15, 2020 (Fig 1A). A total of 315 children were screened and consented, and 246 were enrolled and randomized. Overall, 53.3% (*n* = 131) were boys; the mean age was 4.6 ± 2.5 years; 83.7% (*n* = 205) reported receiving routine sickle cell care; 69.0% (*n* = 169) reported being treated for malaria in the prior 12 months; and the mean hemoglobin concentration was 7.9 ± 1.2 g/dL. Baseline characteristics were similar across the 3 groups (Table 1). Between October 2018 and May 2019, the DSMB temporarily suspended administration of SP-AQ owing to the observation that the first 4 participant deaths occurred among children receiving SP-AQ. Pending the implementation of enhanced safety monitoring, addition of eligibility criteria, and adoption of a stopping rule,

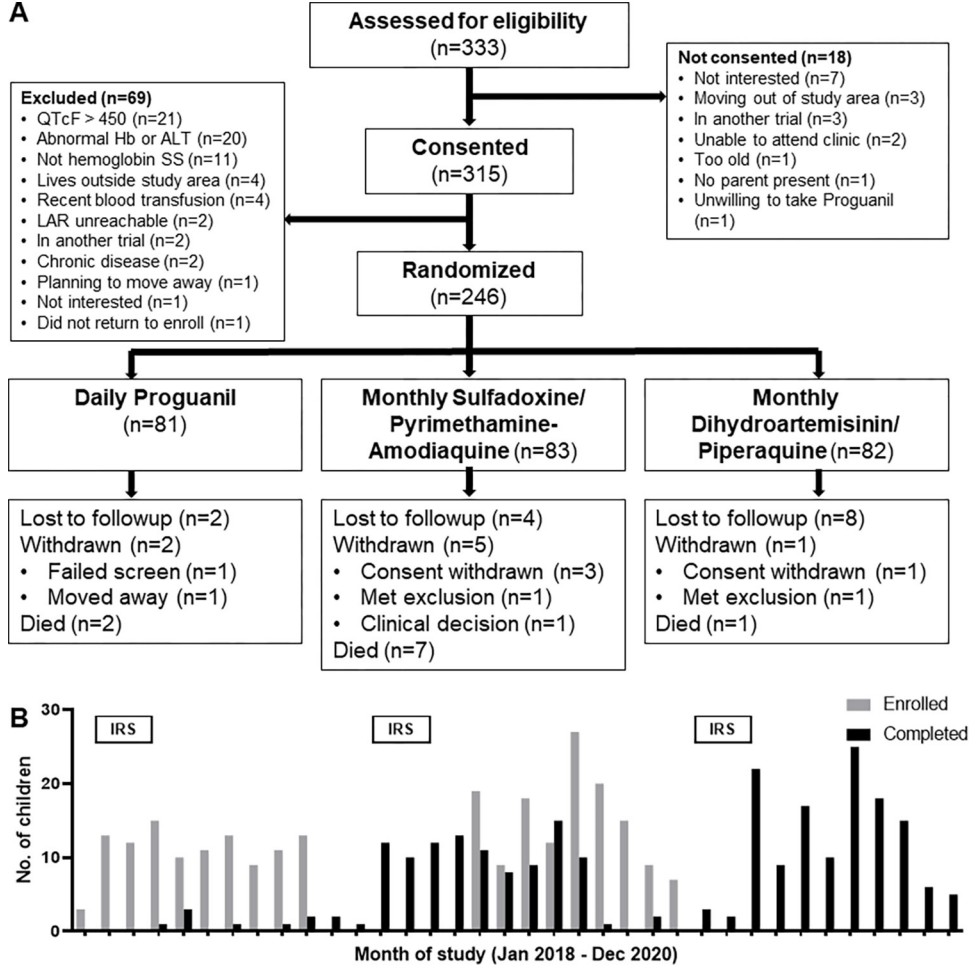

**Fig 1. Screening, randomization, enrollment, and follow-up.** (**A**) Patient flow diagram. (**B**) Monthly enrollment (gray bars) and completion (black bars) of participants. Bars labeled IRS indicate the months in which IRS activities were implemented as malaria control in the study community by health authorities. ALT, alanine aminotransferase; Hb, hemoglobin concentration; IRS, indoor residual spraying; LAR, legally authorized representative; QTcF, QT interval corrected for heart rate using Fridericia's method.

**Table 1. Characteristics of the patients at randomization.**

| | All patients (N = 246) | Daily Proguanil (N = 81) | Monthly SP-AQ (N = 83) | Monthly DP (N = 82) |
|---|---|---|---|---|
| **Demographics** | | | | |
| Boys, % (n/N) | 53.3 (131/246) | 45.7 (37/81) | 56.6 (47/83) | 57.3 (47/82) |
| Luo, % (n/N) | 99.2 (244/246) | 98.8 (80/81) | 98.8 (82/83) | 100 (82/82) |
| Age, year, mean (SD) | 4.6 (2.5) | 4.8 (2.6) | 4.3 (2.3) | 4.7 (2.6) |
| Mid-upper arm circumference, cm, mean (SD)* | 15.1 (1.2) | 15.2 (1.2) | 15.0 (1.1) | 15.1 (1.3) |
| **Medical history** | | | | |
| Receiving regular sickle cell care, % (n/N) | 83.7 (205/245) | 79.0 (64/81) | 90.2 (74/82) | 81.7 (67/82) |
| Routine medications, % (n/N) | | | | |
| Proguanil | 93.1 (229/246) | 90.1 (73/81) | 95.2 (79/83) | 93.9 (77/82) |
| Folic acid | 93.9 (231/246) | 90.1 (73/81) | 96.4 (80/83) | 95.1 (78/82) |
| Penicillin* | 81.6 (102/125) | 79.5 (31/39) | 81.4 (35/43) | 83.7 (36/43) |
| Hydroxyurea | 45.5 (112/246) | 51.9 (42/81) | 44.6 (37/83) | 40.2 (33/82) |
| Nights slept under an ITN in prior week, mean (SD) | 6.9 (0.9) | 7 (0.3) | 6.7 (1.2) | 6.9 (0.8) |
| House treated with IRS in prior 6 months, % (n/N) | 26.6 (65/244) | 33.3 (27/81) | 21.0 (17/81) | 25.6 (21/82) |
| Treated for malaria in prior 12 months, % (n/N) | 69.0 (169/245) | 71.6 (58/81) | 65.9 (54/82) | 69.5 (57/82) |
| Hospitalizations in prior 12 months, % (n/N) | | | | |
| None | 50.2 (123/245) | 51.9 (42/81) | 52.4 (43/82) | 46.3 (38/82) |
| 1–2 | 43.3 (106/245) | 39.5 (32/81) | 42.7 (35/82) | 47.6 (39/82) |
| 3–10 | 6.1 (15/245) | 7.4 (6/81) | 4.9 (4/82) | 6.1 (5/82) |
| >10 | 0.4 (1/245) | 1.2 (1/81) | 0 | 0 |
| Pain crises in prior 12 months, % (n/N) | | | | |
| None | 15.1 (37/245) | 16.0 (13/81) | 15.9 (13/82) | 13.4 (11/82) |
| 1–2 | 30.2 (74/245) | 27.2 (22/81) | 34.1 (28/82) | 29.3 (24/82) |
| 3–10 | 42.9 (105/245) | 44.4 (36/81) | 41.5 (34/82) | 42.7 (35/82) |
| >10 | 11.8 (29/245) | 12.3 (10/81) | 8.5 (7/82) | 14.6 (12/82) |
| Dactylitis episodes in prior 12 months, % (n/N) | | | | |
| None | 51.0 (125/245) | 50.6 (41/81) | 50.0 (41/82) | 52.4 (43/82) |
| Once | 11.4 (28/245) | 16.0 (13/81) | 7.3 (6/82) | 11.0 (9/82) |
| Twice | 11.0 (27/245) | 11.1 (9/81) | 12.2 (10/82) | 9.8 (8/82) |
| 3–5 | 14.3 (35/245) | 16.0 (13/81) | 17.1 (14/82) | 9.8 (8/82) |
| 6–10 | 5.7 (14/245) | 1.2 (1/81) | 7.3 (6/82) | 8.5 (7/82) |
| >10 | 6.5 (16/245) | 4.9 (4/81) | 6.1 (5/82) | 8.5 (7/82) |
| **Laboratory measures** | | | | |
| Hemoglobin, g/dL, mean (SD)** | 7.9 (1.2) | 7.9 (1.1) | 7.6 (1.1) | 8.0 (1.2) |
| % Hemoglobin F, median (Q1, Q3) | 9.9 (4.4, 17.7) | 9.3 (5.1, 17.7) | 9.7 (4.4, 16.7) | 12.0 (4.4, 19.3) |
| ALT, U/L, mean (SD)** | 20.9 (8.7) | 21.4 (9.7) | 20.8 (8.3) | 20.5 (8.3) |
| sCr, umol/L, mean (SD)** | 21.5 (6.7) | 22.3 (6.6) | 20.7 (6.4) | 21.6 (7.2) |
| QTcF interval, mean (range) | 427 (308, 450) | 430 (386, 450) | 424 (308, 448) | 426 (369, 450) |

*Only in those <5 years.

**Measured either during screening or at enrollment visit.

ALT, alanine aminotransferase; DP, dihydroartemisinin-piperaquine; g/dL, grams per deciliter; IRS, indoor residual spraying; ITN, insecticide-treated bed net; QTcF, QT interval corrected for heart rate using Fridericia's method; Q1, Q3, 25th and 75th percentiles; sCr, serum creatinine; SD, standard deviation; SP-AQ, sulfadoxine-pyrimethamine-amodiaquine; U/L, units per liter.

the SP-AQ group temporarily crossed over to Proguanil; SP-AQ dispensing resumed in May 2019. Overall, of 83 children randomized to SP-AQ, 31 (37.3%) crossed to Proguanil, a mean of 155 ± 82 days after randomization, and 17 of these (54.8%) remained on study in May 2019 and subsequently restarted SP-AQ. As a result of this unexpected protocol change, primary efficacy analyses were performed on the AT population, defined as the population of all participants with treatment assignment reflecting the actual treatment received during the month under observation, and safety analyses performed on the ATP population. Overall 8.1% (*n* = 20) of children were withdrawn or lost to follow-up.

Owing to substantial local malaria transmission, county authorities began a campaign in February 2018 of indoor residual spraying with primiphos-methyl. Each yearly campaign lasted 6 weeks and was reinitiated in January 2019 and February 2020 (**Fig 1B**).

Adherence to prescribed chemoprevention was overall high (**Table A in S1 Text**), although more guardians reported difficulty getting children to take SP-AQ (27.5%) than Proguanil (3.7%) or DP (6.3%) and forgetting to administer Proguanil (16%) than SP-AQ (2.5%) or DP (0%).

## Efficacy

We recorded 9 primary outcome malaria events overall, yielding a clinical malaria event IR of 0.04 events/PPY at risk (**Table 2**). In the AT population, the rates were not significantly different to the Proguanil group (0.03/PPY) in both the recipients of SP-AQ (0.09/PPY; IRR: 3.05; 95% CI: 0.36 to 26.14; *p* = 0.39) and DP (0.04/PPY; IRR: 1.36; 95% CI: 0.21 to 8.85; *p* = 0.90). Similar malaria rates were observed in the ITT population (**Table 2**), and when comparing DP to SP-AQ recipients (**Table B in S1 Text**).

We recorded 209 self-reported painful events, yielding a rate of 4.30 events/PPY at risk (**Table 2**). In the AT population, compared to the rate of painful events in the Proguanil group (4.42/PPY), painful events were not significantly different in the recipients of SP-AQ (4.28/PPY; IRR: 0.97; 95% CI: 0.70 to 1.35; *p* = 0.96) and DP (4.18/PPY; IRR: 0.95; 95% CI: 0.65 to 1.38; *p* = 0.92). Similar rates and differences were observed in the ITT population, and when comparing DP to SP-AQ recipients.

Among secondary parasitologic outcomes, DP reduced asymptomatic parasitization by *P. falciparum* by 79% (IRR: 0.21; 95% CI: 0.08 to 0.56; *p* = 0.002). Most other secondary parasitologic outcomes were rare and did not vary significantly between chemoprevention groups (**Table 3**); results in the ITT population were similar (**Table C in S1 Text**). Among secondary hematologic outcomes, we recorded 81 episodes of dactylitis (0.84 episodes/PPY), and the incidence of dactylitis was reduced relative to that in Proguanil recipients (1.08 episodes/PPY) in the recipients of DP (0.51/PPY; IRR: 0.47; 95% CI: 0.23 to 0.96; *p* = 0.038). Compared to DP recipients, SP-AQ recipients had similar incidences of asymptomatic *P. falciparum* parasitization (IRR 2.23; 95% CI: 0.68 to 7.29; *p* = 0.23) and dactylitis (IRR 1.66; 95% CI: 0.77 to 3.56; *p* = 0.24) (**Table B**).

In the AT population analyses of prespecified subgroups on main hematologic outcomes, rates of painful events did not vary between chemoprevention recipients within groups (**Fig 2**, **Table D in S1 Text**). These subgroups also did not modify the association of DP with reduced rates of dactylitis, which was observed irrespective of age, sex, baseline hemoglobin, or use of hydroxyurea.

## Safety

There were 102 SAEs overall in the AT population, in whom 41.9% (90/215) experienced at least 1 SAE, yielding an IR of 0.91 SAEs/PPY at risk. The incidence of SAEs was similar in recipients of Proguanil (0.90 events/PPY; 95% CI: 0.64 to 1.26), SP-AQ (1.10/PPY; 95% CI:

**Table 2. Main parasitologic and hematologic outcomes.**

| | Overall (n = 246) | Daily Proguanil (n = 81) | Monthly SP-AQ (n = 83) | Monthly DP (n = 82) |
|---|---|---|---|---|
| **AT population** | | | | |
| **Patient-years follow-up** | **208.52** | **82.5** | **56.7** | **69.3** |
| **Clinical malaria** | | | | |
| Number of events | 9 | 3 | 3 | 3 |
| Incidence rate per patient-year (95% CI) | 0.04 (0.02–0.08) | 0.03 (0.01–0.09) | 0.09 (0.02–0.39) | 0.04 (0.01–0.13) |
| Incidence rate ratio (95% CI)* | | | 3.05 (0.36–26.14) | 1.36 (0.21–8.85) |
| p-value | | | 0.39 | 0.90 |
| **Painful events** | | | | |
| Number of events | 897 | 373 | 238 | 286 |
| Incidence rate per patient-year (95% CI) | 4.30 (3.77–4.2) | 4.42 (3.63–5.38) | 4.28 (3.38–5.42) | 4.18 (3.19–5.46) |
| Incidence rate ratio (95% CI)* | | | 0.97 (0.70–1.35) | 0.95 (0.65–1.38) |
| p-value | | | 0.96 | 0.92 |
| **ITT population** | | | | |
| **Patient-years follow-up** | **208.5** | **71.6** | **67.6** | **69.3** |
| **Clinical malaria** | | | | |
| Number of events | 9 | 3 | 3 | 3 |
| Incidence rate per patient-year (95% CI) | 0.04 (0.02–0.08) | 0.04 (0.01–0.13) | 0.04 (0.01–0.14) | 0.04 (0.01–0.13) |
| Incidence rate ratio (95% CI)* | | | 1.06 (0.18–6.28) | 1.03 (0.17–6.12) |
| p-value | | | >0.99 | >0.99 |
| **Painful events** | | | | |
| Number of events | 897 | 339 | 272 | 286 |
| Incidence rate per patient-year (95% CI) | 4.36 (3.81–4.98) | 4.77 (3.85–5.92) | 4.11 (3.29–5.13) | 4.17 (3.22–5.40) |
| Incidence rate ratio (95% CI)* | | | 0.86 (0.61–1.22) | 0.87 (0.60–1.28) |
| p-value | | | 0.54 | 0.65 |

*Relative to daily Proguanil.

AT, As-Treated; CI, confidence interval; DP, dihydroartemisinin-piperaquine; ITT, Intention-To-Treat; SP-AQ, sulfadoxine/pyrimethamine-amodiaquine.

0.78 to 1.54), and DP (0.82/PPY; 95% CI: 0.54 to 1.24) (**Table 4**). The most commonly reported SAEs were hospitalizations resulting from a painful crisis (91 events), from anemia (33 events), and malaria or sepsis (12 events each) (**Table E in S1 Text**). One SAE was judged by the safety monitor to be related to chemoprevention regimen: an ALT elevation to 517 in a recipient of SP-AQ, which normalized following cessation of drug. We recorded 72 all-cause hospitalizations in the AT population, yielding an overall rate of hospitalization of 0.67/PPY at risk (**Table 4**); the prevalence and the rate of hospitalization was similar between groups. Ten children died: 2 receiving daily Proguanil, 7 receiving SP-AQ, and 1 receiving DP (**Table 4**); relative to Proguanil (cumulative IR [CIR] 2.2%), the risk of death was elevated in SP-AQ recipients (CIR 11.3%; HR 5.44; 95% CI 0.92 to 32.11; p = 0.064), although this did not reach statistical significance and was not elevated in DP recipients (CIR 1.3%; HR 0.61; 95% CI 0.04 to 9.22; p = 0.89). Similarly, relative to DP recipients, the risk of death was elevated in SP-AQ

**Table 3. Secondary outcomes in the AT population.**

| | Overall (n = 246) | Daily Proguanil (n = 81) | Monthly SP-AQ (n = 83) | Monthly DP (n = 82) |
|---|---|---|---|---|
| Patient-years follow-up | 208.52 | 82.5 | 56.7 | 69.3 |
| **Parasitologic** | | | | |
| **Severe malaria** | | | | |
| Incidence rate per patient-year (95% CI) | 0 | 0 | 0 | 0 |
| **Hospitalized for malaria** | | | | |
| Number of events | 7 | 2 | 2 | 3 |
| Incidence rate per patient-year (95% CI) | 0.04 (0.02–0.09) | 0.02 (0–0.08) | 0.02 (0.01–0.11) | 0.08 (0.02–0.32) |
| Incidence rate ratio (95% CI)* | | | 1.34 (0.13–13.70) | 4.17 (0.42–41.34) |
| p-value | | | 0.94 | 0.27 |
| **Light microscopy–positive malaria** | | | | |
| Number of events | 47 | 18 | 10 | 19 |
| Incidence rate per patient-year (95% CI) | 0.19 (0.14–0.26) | 0.18 (0.10–0.32) | 0.15 (0.07–0.31) | 0.25 (0.15–0.42) |
| Incidence rate ratio (95% CI)* | | | 0.80 (0.28–2.32) | 1.35 (0.56–3.27) |
| p-value | | | 0.85 | 0.65 |
| **Unconfirmed malaria** | | | | |
| Number of events | 107 | 46 | 21 | 40 |
| Incidence rate per patient-year (95% CI) | 0.50 (0.39–0.65) | 0.49 (0.33–0.71) | 0.45 (0.26–0.81) | 0.57 (0.40–0.82) |
| Incidence rate ratio (95% CI)* | | | 0.93 (0.45–1.96) | 1.18 (0.65–2.13) |
| p-value | | | 0.97 | 0.76 |
| **Fatal malaria** | | | | |
| Cumulative incidence rate (95% CI) | 0 | 0 | 0 | 0 |
| **Asymptomatic *P. falciparum* infection** | | | | |
| Number of events | 74 | 50 | 15 | 9 |
| Incidence rate per patient-year (95% CI) | 0.35 (0.24–0.50) | 0.58 (0.36–0.96) | 0.27 (0.14–0.52) | 0.12 (0.06–0.25) |
| Incidence rate ratio (95% CI)* | | | 0.46 (0.17–1.20) | 0.21 (0.08–0.56) |
| p-value | | | 0.13 | 0.002 |
| **Hematologic** | | | | |
| **Dactylitis** | | | | |
| Number of events count | 180 | 98 | 45 | 37 |
| Incidence rate per patient-year (95% CI) | 0.84 (0.62–1.13) | 1.08 (0.68–1.71) | 0.84 (0.53–1.35) | 0.51 (0.33–0.78) |
| Incidence rate ratio (95% CI)* | | | 0.78 (0.41–1.50) | 0.47 (0.23–0.96) |
| p-value | | | 0.60 | 0.04 |
| **Severe anemia*** | | | | |
| Number of events | 21 | 9 | 8 | 4 |
| Incidence rate per patient-year (95% CI) | 0.10 (0.07–0.15) | 0.11 (0.06–0.21) | 0.14 (0.07–0.29) | 0.06 (0.02–0.16) |
| Incidence rate ratio (95% CI)* | | | 1.30 (0.43–3.91) | 0.53 (0.14–2.07) |
| p-value | | | 0.81 | 0.47 |

(*Continued*)

**Table 3.** (Continued)

|  | Overall (*n* = 246) | Daily Proguanil (*n* = 81) | Monthly SP-AQ (*n* = 83) | Monthly DP (*n* = 82) |
|---|---|---|---|---|
| **Transfusion of blood products** |  |  |  |  |
| Number of events | 24 | 15 | 5 | 4 |
| Incidence rate per patient-year (95% CI) | 0.15 (0.08–0.31) | 0.18 (0.07–0.44) | 0.09 (0.03–0.30) | 0.19 (0.04–0.99) |
| Incidence rate ratio (95% CI)* |  |  | 0.52 (0.10–2.79) | 1.08 (0.13–9.19) |
| *p*-value |  |  | 0.58 | >0.99 |
| **Acute chest syndrome** |  |  |  |  |
| Number of events | 4 | 1 | 3 | 0 |
| Incidence rate per patient-year (95% CI) | 0.01 (0–0.05) | 0.01 (0–0.09) | 0.05 (0.01–0.20) | 0 |
| Incidence rate ratio (95% CI)* |  |  | 3.96 (0.27–57.42) |  |
| *p*-value |  |  | 0.37 |  |
| **General** |  |  |  |  |
| **All-cause hospitalization** |  |  |  |  |
| Number of events | 139 | 56 | 43 | 40 |
| Incidence rate per patient-year (95% CI) | 0.67 (0.54–0.83) | 0.70 (0.49–0.99) | 1.28 (0.54–2.99) | 0.78 (0.44–1.37) |
| Incidence rate ratio (95% CI)* |  |  | 1.83 (0.63–5.31) | 1.11 (0.52–2.39) |
| *p*-value |  |  | 0.35 | 0.93 |
| **Death** |  |  |  |  |
| Cumulative incidence | 10 | 2 | 7 | 1 |
| Cumulative incidence rate (95% CI) | 4.4% (2.4–8.1) | 2.2% (0.5–9.3) | 11.3% (6.1–20.7) | 1.3% (0.2–9.1) |
| Hazard ratio (95% CI)* |  |  | 5.44 (0.92–32.11) | 0.61 (0.04–9.22) |
| *p*-value |  |  | 0.064 | 0.89 |

*Defined as hemoglobin concentration <5.5 g/dL.

AT, As-Treated; CI, confidence interval; DP, dihydroartemisinin-piperaquine; g/dL, grams per deciliter; SP-AQ, sulfadoxine/pyrimethamine-amodiaquine.

recipients (HR 8.86; 95% CI: 0.74 to 106.55; *p* = 0.10), although this did not achieve statistical significance (**Table B in S1 Text**).

Among laboratory events, neither the change in hemoglobin concentration from baseline nor the incidence of severe anemia, defined as a drop of more than 2 g/dL from baseline, differed between chemoprevention groups (**Table 4**). Similarly, there were no differences between groups in the change in baseline nor the incidence of elevated values of ALT or of sCr. Follow-up neutrophil and platelet counts in the ITT population are presented in **Tables F and G in S1 Text**. Among the 7 children who died while receiving SP-AQ, 2 were G6PD deficient (2 hemizygote boys), and 1 was heterozygote for a CYP2C8*2 allele associated with reduced metabolism of AQ (**Table H in S1 Text**).

Among DP recipients, we enrolled 12 children in a substudy of QTcF measurements following the third dose of their monthly courses of DP. Among 134 follow-up QTcF measurements in these children, we recorded 16 (11.9%) measurements of QTcF >450 ms, which occurred in 50% (6/12) of children, with a maximum QTcF of 479 ms (**Figure A in S1 Text**). There was 1 (0.7%) case of a prolongation of >50 ms QTcF compared to baseline to 52 ms

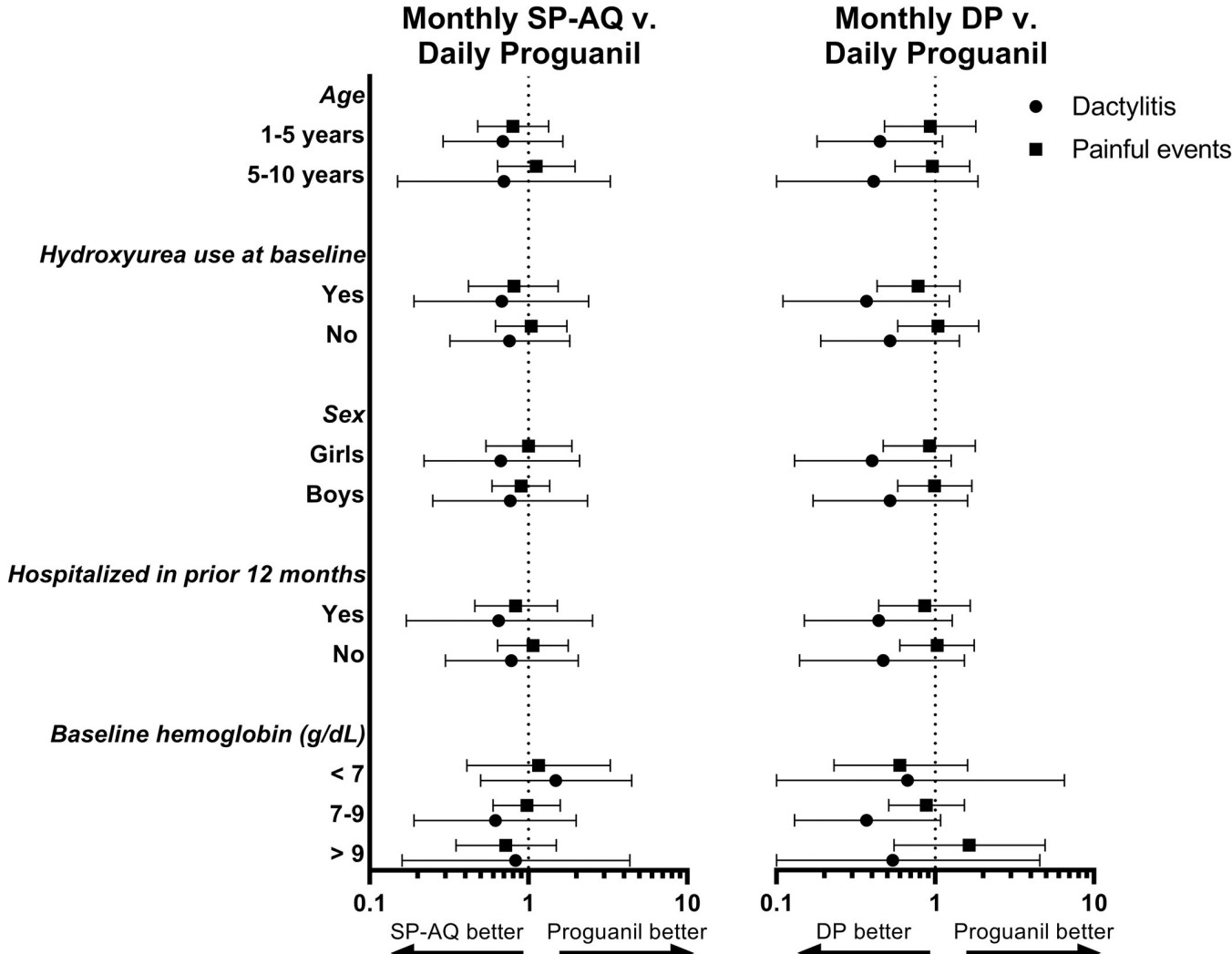

**Fig 2. Main hematologic outcomes in the AT population according to subgroup.** Points are estimates of the IRR relative to daily Proguanil. Bars are 95% CI. AT, As-Treated; CI, confidence interval; DP, dihydroartemisinin-piperaquine; g/dL, grams per deciliter; IRR, incidence rate ratio; SP-AQ, sulfadoxine/pyrimethamine-amodiaquine.

(**Table 4, Table I in S1 Text**). Mean (SD) QTcF changes from baseline peaked at 17 (13) ms at 6 months and then declined to 8 (16) ms after 12 months of administration.

## Discussion

In this randomized, open-label trial, we compared 2 monthly malaria chemoprevention regimens with the standard-of-care of daily Proguanil in Kenyan children with SCA in an historically malaria-holoendemic setting in Western Kenya. Consistent with enhanced community malaria control that sharply reduced transmission, we recorded very few episodes of malaria. Compared to Proguanil, neither monthly SP-AQ nor DP reduced the incidence of the primary outcome clinical malaria, although monthly DP did significantly reduce the incidence of asymptomatic parasitization. Hematologic outcomes were common, and we observed a significant reduction in dactylitis in the children receiving monthly DP. Given the observed safety of DP, the high adherence to it in our participants, and its known high efficacy as malaria

**Table 4. Safety outcomes and SAEs in ATP population.**

| | Overall (*n* = 215) | Daily Proguanil (*n* = 81) | Monthly SP-AQ (*n* = 52) | Monthly DP (*n* = 82) |
|---|---|---|---|---|
| **General outcomes** | | | | |
| All-cause hospitalizations, % (n) | 33.5 (72) | 34.6 (28) | 36.5 (19) | 30.5 (25) |
| Hospitalization incidence, events/PPY (95% CI) | 0.68 (0.54–0.86) | 0.66 (0.46–0.94) | 0.85 (0.59–1.23) | 0.60 (0.38–0.95) |
| Study drug withdrawn, % (n) | 0.5 (1) | 0 | 1.9 (1) | 0 |
| All-cause deaths, % (n)** | 4.2 (9) | 2.5 (2) | 11.5 (6) | 1.2 (1) |
| Any SAE, % (n) | 41.9 (90) | 44.4 (36) | 48.1 (25) | 35.4 (29) |
| SAE incidence, events/PPY (95% CI) | 0.91 (0.74–1.13) | 0.90 (0.64–1.26) | 1.10 (0.78–1.54) | 0.82 (0.54–1.24) |
| **Laboratory outcomes** | | | | |
| Change in Hb from baseline to 12 months, mean (range), g/dL | 0.2 (−2.5, 3.6) | 0 (−2.5, 2.2) | 0.4 (−2.1, 3.6) | 0.3 (−1.8, 3.5) |
| Drop in Hb >2 g/dL from baseline, % (n) | 6.1 (13) | 3.7 (3) | 3.9 (2) | 9.8 (8) |
| Change in creatinine from baseline to 12 months, mean (range), umol/L | 0.78 (−25.7, 19.4) | −0.31 (−25.7, 18.6) | 0.50 (−13.6, 13.1) | 2.05 (−18.8, 19.4) |
| sCr >62, % (n) | 0.5 (1) | 0 | 2.0 (1) | 0 |
| Change in ALT from baseline to 12 months, mean (range), U/L | 5.39 (−66.2, 605.1) | 8.64 (−66.2, 605.1) | 4.38 (−23.4, 72.5) | 2.55 (−37.5, 97.4) |
| Serum ALT >60, % (n) | 7.5 (16) | 8.6 (7) | 5.9 (3) | 7.3 (6) |
| **ECG outcomes*** | | | | |
| Any QTcF >450 ms, % (n/N) | NA | NA | NA | 50.0 (6/12) |
| Any QTcF change from baseline >50 ms, % (n/N)* | NA | NA | NA | 8.3 (1/12) |

*Among only monthly DP recipients enrolled in the monthly ECG monitoring substudy.

**One additional participant allocated to SP-AQ died but does not appear in the ATP population owing to their crossover to the Proguanil arm.

ALT, alanine aminotransferase; ATP, According-to-Protocol; CI, confidence interval; DP, dihydroartemisinin-piperaquine; ECG, electrocardiogram; Hb, hemoglobin; ms, millisecond; PPY, per patient-year; QTcF, QT interval corrected for heart rate using Fridericia's method; SAE, serious adverse event; sCr, serum creatinine; SP-AQ, sulfadoxine/pyrimethamine-amodiaquine; U/L, units per liter.

prevention, monthly DP is a promising and feasible alternative as malaria chemoprevention in children with SCA to prevent both parasitologic and hematologic morbidity.

Monthly DP was associated with a reduced rate of episodes of dactylitis compared to daily Proguanil, both overall and in each examined subgroup. The prophylactic combination of penicillin and chloroquine reduced dactylitis in an early Ugandan trial [25], contributing to current recommendations for malaria chemoprophylaxis [11]. Monthly DP is highly effective as malaria prevention in children and in pregnant women owing to its immediate parasiticidal effect as well as the prolonged half-life of the piperaquine component, and, consistent with this, we observed a nearly 80% reduction in asymptomatic *P. falciparum* infections in DP recipients. Notably, this effect of DP on dactylitis was not observed on painful events, despite the more than 5-fold increase in the overall incidence of painful events and presumably similar pathophysiology. Other measured hematologic events were too rare to capture differences between groups, and we did not observe a difference in the change in hemoglobin concentration between groups. Whether a measurable effect of malaria chemoprophylaxis on common hematologic outcomes like painful crises can be captured in a higher-transmission setting remains an open question.

We observed for DP high adherence and acceptability by children (**Table A in S1 Text**). We recorded no drug-related SAEs for DP, and the scope and incidence of SAEs were similar to that for SP-AQ and Proguanil. As expected, we observed a prolongation of the QTcF

interval following the third dose after repeated courses in a small subset of recipients. Although half of monitored children had at least 1 QTcF exceeding 450 ms, in only 1 recording was the QTcF prolonged beyond 50 ms from baseline, and no recipient required DP alteration. It is notable that 21 children failed screening owing to QTcF >450 ms, highlighting that baseline QT prolongation is common in these children. This may merit baseline QT measurement to limit DP-induced prolongation, although this risk may be mitigated by alternate DP dosing schemes or by, as we observed, the plateauing of QT intervals after 6 months of prophylaxis (**Table I and Figure A in S1 Text**). These clinical factors would need to be complemented before wider policy adoption by considerations of DP resistance, which was detected as declining clinical efficacy first in Southeast Asia [26] and more recently at several sites in Africa [27,28]. Although wider use of DP will necessarily increase pressure to develop resistance, this may be mitigated by a restriction of DP use to high-risk populations like pregnant women and children with SCA, and by the observation through multiple studies [29–32] that mass drug administration with DP across diverse populations has not reliably increased molecular markers of DP resistance.

On balance of efficacy and safety, monthly SP-AQ does not appear to have a role as chemoprevention in children with SCA. Reductions in asymptomatic *P. falciparum* infection and dactylitis did not reach statistical significance, and while these were undermined by low malaria transmission and likely by prevalent resistance to SP in Kenya [33], rates of general outcomes did not suggest benefit of the putative antibacterial activity of the sulfadoxine component. More notable is the severity of SAEs with SP-AQ. Seven children receiving SP-AQ died, and 1 additional child required cessation owing to transaminitis. Causes of death in children receiving SP-AQ were variable, including severe anemia, acute chest syndrome, and painful crisis, which were among the most common SAEs overall, and the durations of SP-AQ treatment prior to death varied from 81 to 294 days (**Table H in S1 Text**). We investigated as a contributing factor the possibility of G6PD-linked anemia owing to the known risk following SP exposure [34,35], but the A- allele of G6PD deficiency was present in only 2 of these 7 children. This high rate of SAEs was unexpected given the favorable safety profile of monthly SP-AQ in recipients as Seasonal Malaria Chemoprophylaxis [36], during which 3 to 4 monthly courses are administered to all children under 5 years of age. This difference could be the result of our use of SP-AQ in a chronically ill population or accumulated risk following repeated monthly dosing.

Our trial had several limitations. As noted above, the low incidence of *P. falciparum* infection prevents meaningful assessment of the primary outcome. Given the plausible link between infection and hematologic outcomes, this may have also limited our ability to fully assess other outcomes. The temporary cross-over of SP-AQ recipients to Proguanil necessitated analytic changes and limited approaches to counting outcomes, and this was mitigated by prioritizing the AT population. Finally, generalizability may be limited by the common baseline use of hydroxyurea and, as a result of trial participation, the enhanced supportive care of participants. Pragmatic studies of prophylaxis delivered through routine sickle cell care can measure effectiveness in diverse care and transmission settings.

In summary, among Kenyan children under 10 years with SCA, monthly DP was very acceptable and reduced the incidence of dactylitis and asymptomatic *P. falciparum* infection despite very limited malaria transmission or clinical malaria. Monthly SP-AQ had no measurable benefit compared to daily Proguanil, and we observed a greater number of deaths in this group, although mortality differences were not statistically significant. Our results provide a rationale to consider the wider use of DP with appropriate monitoring as a routine component of care for children with SCA in malaria-endemic settings.

## Supporting information

**S1 CONSORT Checklist. CONSORT checklist.**
(DOC)

**S1 Protocol. Study protocol.**
(DOCX)

**S1 Analysis. Statistical analysis plan.**
(DOCX)

**S1 Text. Supporting information.** Supplemental methods, figure, and tables.
(DOCX)

## Acknowledgments

We thank the children and their parents for their participation; the site staff Ernest Ojwang, Seline Miruni, Faith Ogolla, and Erick Ayaye; the trial pharmacists Collins Saina and Linet Kugo; data managers Steven Karuru and Edna Sang; the members of the data and safety monitoring board (David Ayuku, Walter Dzik, Gregory Kato, Karen Kessler, Jennifer Knight-Madden, Irene Marete, Rebecca Pentz, Jennifer Rothman, and Liz Turner); the members of our Community Advisory Board; Stanley Odanga and Terry Odero for community engagement; Stacey Gondi for external trial monitoring; John Humphrey for independent safety monitoring; the CEOs of Homa Bay County Hospital Drs. Lillian Kochola and Meshack Liru for their accommodation of trial activities; Gayle Passmore, Minal Bhojai, Varsha Gajjar, Casey Silver, Lucy Abel, George Ambani, Tabitha Jepkurgat, Debbie Drosdick, and Assumpta Nantume for their operational assistance; Laura Edwards and Sharon Stroud for statistical programming support; the AMPATH Reference Laboratory for assay support; and Francis Kithuku, Dominique Cole, Robert Rono, and Benta Kamire for their administrative support.

## Author Contributions

**Conceptualization:** Steve M. Taylor, Wendy P. O'Meara, Festus M. Njuguna.

**Formal analysis:** Steve M. Taylor, Angie Wu, Cynthia L. Green.

**Funding acquisition:** Steve M. Taylor, Wendy P. O'Meara, Festus M. Njuguna.

**Investigation:** Steve M. Taylor, Sarah Korwa, Betsy Freedman, Wendy P. O'Meara, Festus M. Njuguna.

**Methodology:** Steve M. Taylor, Betsy Freedman.

**Project administration:** Steve M. Taylor, Sarah Korwa, Sheila Clapp, Joseph Kipkoech Kirui, Wendy P. O'Meara, Festus M. Njuguna.

**Supervision:** Steve M. Taylor, Cynthia L. Green, Sheila Clapp, Joseph Kipkoech Kirui, Festus M. Njuguna.

**Writing – original draft:** Steve M. Taylor.

**Writing – review & editing:** Angie Wu, Cynthia L. Green, Betsy Freedman, Joseph Kipkoech Kirui, Wendy P. O'Meara, Festus M. Njuguna.

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
