## [Editor Report · Decision Letter 0]

16 Mar 2022

Dear Dr Taylor, 

Thank you for submitting your manuscript entitled "Monthly sulfadoxine/pyrimethamine-amodiaquine or dihydroartemisinin-piperaquine as malaria chemoprevention in young children with sickle cell anemia: A randomized controlled trial" for consideration by PLOS Medicine.

Your manuscript has now been evaluated by the PLOS Medicine editorial staff and I am writing to let you know that we would like to send your submission out for external assessment.

However, we first need you to complete your submission by providing the metadata that is required for full assessment. To this end, please login to Editorial Manager where you will find the paper in the 'Submissions Needing Revisions' folder on your homepage. Please click 'Revise Submission' from the Action Links and complete all additional questions in the submission questionnaire.

Please re-submit your manuscript within two working days, i.e. by Mar 18 2022 11:59PM.

Once your full submission is complete, your paper will undergo a series of checks in preparation for external assessment. 

Kind regards,

Richard Turner, PhD

rturner@plos.org

---

## [Decision Letter · Decision Letter 1]

12 Apr 2022

Dear Dr. Taylor,

Thank you very much for submitting your manuscript "Monthly sulfadoxine/pyrimethamine-amodiaquine or dihydroartemisinin-piperaquine as malaria chemoprevention in young children with sickle cell anemia: A randomized controlled trial" (PMEDICINE-D-22-00862R1) for consideration at PLOS Medicine. 

Your paper was discussed with an academic editor with relevant expertise and sent to independent reviewers, including a statistical reviewer. The reviews are appended at the bottom of this email and any accompanying reviewer attachments can be seen via the link below:

[LINK]

In light of these reviews, we will not be able to accept the manuscript for publication in the journal in its current form, but we would like to invite you to submit a revised version that addresses the reviewers' and editors' comments fully. You will appreciate that we cannot make a decision about publication until we have seen the revised manuscript and your response, and we expect to seek re-review by one or more of the reviewers. 

We hope to receive your revised manuscript by May 03 2022 11:59PM. Please email us (plosmedicine@plos.org) if you have any questions or concerns.

Please let me know if you have any questions, and we look forward to receiving your revised manuscript. 

Sincerely,

Richard Turner, PhD

Senior editor, PLOS Medicine

rturner@plos.org

Please adapt the data statement (submission form) as appropriate, noting that authors cannot be listed as contacts for inquiries about data access according to PLOS' policy (https://journals.plos.org/plosmedicine/s/data-availability).

Please quote the country name in your title. 

Please quote the study dates in your abstract. 

Please quote participant demographic details in the abstract.

Noting "p<0.01" at line 54 (abstract), please quote exact p values or "p<0.001" throughout.

Please add a new final sentence to the "Methods and findings" subsection of your abstract, which should begin "Study limitations include ..." or similar and should quote 2-3 of the study's main limitations. 

After the abstract, please add a new and accessible "Author summary" section in non-identical prose. You may find it helpful to consult one or two recent research papers in PLOS Medicine to get a sense of the preferred style. 

Throughout the main text, please style reference call-outs as follows: " ... complications [6,7]." (noting the absence of spaces within the square brackets). 

Noting reference 1 and any others, please use the journal name abbreviation "PLoS Med."; and "PLoS ONE" where appropriate.

Noting references 8 & 13, we suggest spelling out the institution names. 

Noting reference 14 and others, please list 6 author names where appropriate, followed by "et al.".

Noting reference 17, please list the individual or institutional author first.

Please add an author name to reference 23.

Please add a completed CONSORT checklist, labelled "S1_CONSORT_Checklist" or similar and referred to as such in the Methods section (main text). 

In the checklist, please refer to individual items by section (e.g., "Methods") and paragraph number, not by page or line numbers as these generally change in the event of publication. 

Comments from academic editor:

The problem of underpowered malaria trials in the presence of malaria elimination efforts is no longer unusual. The efficacy of antimalarial prophylaxis against asymptomatic infections becomes a useful surrogate marker in such circumstances. The impact of the 3 drug regimens on asymptomatic infections should be included in the abstract. The authors could also create an additional endpoint by combining asymptomatic and symptomatic Pf infections to increase the power. The incidence rate of asymptomatic infections in children receiving proguanil (0.58/year)is halved by prophylaxis with SP-AQ (0.27/year) and this rate is further halved by DP (0.12/year) – table 3. Similarly, dactylitis decrease from 1.08/year in patients treated with proguanil to 0.84/yea in patients treated with SP-AQ and 0.51/year in patients in DP. There is an intriguing trend suggesting that SP-AQ is more beneficial than proguanil but less beneficial than DP. The authors conclude that there is no role for SP-AQ and DP is the solution. This may be jumping to conclusions as Reviewer 2 points out. SP-AQ has by now been used in 10s of millions of children on seasonal chemoprophylaxis we know it is safe, available and accepted. DP may be ideal, but policymakers may be reluctant to roll out DP for prophylaxis. The experience in Cambodia shows that DP -resistance is a real danger. If DP no longer works, there are few options left. This threat of emerging and spreading resistance should be discussed by authors.

The authors describe and discuss safety issues extensively but not in a helpful fashion. The excess mortality in the SP-AQ arm is not statistically significant this is most probably due to chance. The authors have to be careful not to mislead the reader. Similarly, the observed variability in QTc intervals sounds worrying the way it is described in the current text and was immediately picked up by reviewer 4. It is most unfortunate that the investigators only measured the QTc in the patients receiving DP. Without a control it is not possible to infer any causality. The variability in QTc has to be analysed much more carefully. At a minimum the authors have to show the QTc variability over the follow-up period for the 12 trial participants on whom they have serial QTcs. Leaving the safety reporting as it is now will create a lot of unnecessary concern.

Comments from the reviewers:

*** Reviewer #1: 

Statistical review

This paper reports a RCT comparing two regimens of chemoprevention vs standard of care for preventing malaria in children with sickle cell anaemia. The trial did not demonstrate differences in the primary endpoint but did find some significant differences in secondary endpoints.

Generally the trial was well-reported and used appropriate statistical methods. I have some minor comments.

1. Abstract: "incidence rates were similar" - I would recommend changing this to "not significantly different" as the confidence intervals allow for fairly large differences in the IRR.

2. Abstract (and throughout results): 'p<0.01' - I would recommend reporting p-values more precisely unless <0.0001.

3. Abstract 'Serious adverse events were common but distributed between groups' - could this be more precisely written that 'distributed between groups', e.g. 'but no arm showed safety concerns'?

4. Abstract: I note the clinicaltrials.gov page has many secondary endpoints that are not mentioned in the abstract. I would recommend (if the abstract reports secondary endpoints) it is clear there were other non-reported secondary endpoints that were not significant.

5. Methods, randomisation: was a fixed block allocation used, or was this stratified or using random block sizes? Providing the block size would be useful.

6. Methods, outcome measures: I would recommend each outcome on the clinicaltrials.gov page is mentioned here and reported in the paper, or a reason for it/them not being reported given here.

7. Methods, statistical analysis: "assuming that each experimental arm would be made" - I would change this to 'each comparison between experimental arm and control would be tested at the 0.0269 level (Dunnett correction)'. 

8. Methods, statistical analysis: I did not quite follow how the as-treated analysis worked. Was the indicator for SP-AQ arm a time varying covariate in the model? Later on it is said the AT population was not evaluated - is this just for secondary outcomes?

9. Methods, statistical analysis: I note the protocol said different count outcome analyses would be considered with the one with the lowest residual variance used - this may be worth briefly mentioning in the analysis methods.

10. Results, lines 238-242: similar comment to comment 1.

James Wason

*** Reviewer #2: 

This trial compared three drug regimens for malaria chemoprevention (prophylaxis) in patient with sickle cell disease in 

western Kenya: daily proguanil (the current recommendation in many countries), monthly SPAQ, and monthly DP.

Unfortunately (from the point of view of the trial aims), there was very little malaria in the trial cohorts, 

9 malaria episodes in total. So there was no power to measure a difference in protection against clinical malaria 

between the drug regimens. There were more episodes of (asymptomatic) malaria infection, but far fewer than the 

sample size calculation envisaged. The main interest will therefore be in the information on tolerability and safety, 

but these need to be more clearly presented. The tables need to be revised, they should include the number of each 

type of event, and the amount of person time. And there is a problem with the analysis approach in that the conclusions 

relate to a comparison of the two monthly regimens, SPAQ and DP, but the statistical analyses are limited to comparison 

of each with the proguanil group. 

The DSMB stopped the SPAQ arm temporarily, patients in that arm of the trial were switched to proguanil, and then 

switched back to SPAQ later, I couldn't see any explanation of the reason for this. 

It will be important to explain clearly how much person time was spent on each treatment and the number of doses received of each type of treatment, 

I didn't see these details.

I felt that the conclusions were not supported by the data. The fact that the analyses didn't directly compare the SPAQ and DP groups may possibly have led to some confusion about how to interpret the results. For example, referring to SPAQ: "There was no signal of efficacy on parasitologic or hematologic outcomes" and "monthly DP did significantly reduce the incidence of asymptomatic parasitization... and we observed a significant 

reduction in dactylitis in the children receiving monthly DP". Such conclusions comparing SPAQ and DP are based on whether or not there was a statistically significant reduction (P<0.05) compared to proguanil, rather than a comparison of the magnitude of reduction between SPAQ and DP. 

A closed testing procedure could be have been used for example. (Using the closed testing procedure, first test the global hypothesis with 

a 2df level α test, and if it is rejected, then test each of pairwise comparison also at level α.) 

It seems unlikely there is any evidence of real differences between SPAQ and DP in this trial. For example, the rate ratio comparing the incidence of dactylis (blocked blood flow in the hands and feet causing painful swelling) in the DP group with the proguanil group was 0.47 (95%CI 0.23 - 0.96), and for the SPAQ group compared to proguanil, 0.78 (95%CI 0.41 - 1.50), it seems unlikely there could be a significant difference between these effects. 

And the rate ratios for asymptomatic malaria of 0.46 (0.17 - 1.20) and 0.21 (0.08 - 0.56), also look similar (the fact that the P-value is significant for DP and not for SPAQ does not indicate there is a difference in magnitude of effect). 

The conclusion that "Monthly SP-AQ does not appear to have a role as chemoprevention in children with sickle cell anemia" does not seem justified on the basis of these data. There were 10 deaths, 7 of them in the SPAQ arm. The hazard ratio comparing the SPAQ group with the proguanil group had a confidence interval from 0.92 to 32. It seems likely that the cause of death was linked to sickle cell disease rather than drug toxicity. Many millions of children, including many thousands with sickle cell disease, receive SPAQ each year in Seasonal Malaria Chemoprevention programmes. An earlier trial of SPAQ for malaria prevention in sickle cell disease (J Infect Dis. 2015 Aug 15;212(4):617-25) did not find any excess mortality - 

that trial, in Nigeria, had 3 arms with a similar number of patients per arm as the current study, there were 7 deaths in total, none of them in the SPAQ group. 

A more minor point related to the statistical analysis is the use of Fine & 203 Gray's method to allow death to be a competing risk, this is questionable. It is better to simply treat all competing events as though the individual were right censored at the time the competing event occurs. See https://statisticalhorizons.com/for-causal-analysis-of-competing-risks/

The sample size section mentioned anticipated incidence rate of malaria of 3.7 episodes per person year but then says the study is powered to detect a reduction of 40% with DP and of 40% with proguanil, as if compared to placebo, but there was no placebo group.

The study is potentially interesting but the analyses and interpretation need to be revised.

*** Reviewer #3: 

In this manuscript Taylor and colleagues report on the efficacy of daily proguanil versus monthly sulfadoxine-pyrimethamine-amodiaquine (SP-AQ) combination versus monthly dihydroartemisinin-piperaquine (DP) for malaria chemoprevention among Kenyan children with sickle cell disease. This is a very important study as it seeks to evaluate alternative drugs for preventing malaria in one of the groups most vulnerable to malaria and its associated morbidity and mortality. Although the authors found no significant difference in the rate of clinical malaria among children who received monthly SP-AQ or monthly DP compared to those who received daily proguanil possibly due to few clinical malaria cases during follow-up because of indoor residual spraying in the study area, monthly DP was safe and was associated with a lower risk of asymptomatic P. falciparum parasitemia and lower rate of dactylitis. Compared to daily proguanil, monthly SP-AQ was associated with a trend towards an increased risk of mortality. This was a well-conducted study but had two main limitations; the lower than expected clinical malaria episodes which reduced the power of the study and the unanticipated crossover where children randomized to SP-AQ received daily proguanil for about 8 months. It would be useful to the readers if the authors mentioned the cause of the cross-over, which in this manuscript is implied as safety concern, but it is not clear.

Other comments

1. Although there was no significant difference in the risk of clinical malaria comparing AS-AQ and DP to daily proguanil, I would suggest that the authors keep this as the lead finding and conclusion and then highlight a few positive related secondary outcomes e.g risk reduction for asymptomatic parasitemia.

2. In the results tables, it would be useful to clearly indicate numerator and denominator numbers used to estimate the incidence/risk, ie number of episodes of clinical malaria and total number of person years of follow-up for each chemoprevention arm. 

Some of the numbers in the tables need to be revised to correct minor errors in rounding off, for example the proportion of households with IRS in table 1 is 26.4%, proportion of children reporting no prior painful crises in the last 12 months is 15.0%. The authors can check all numbers in the tables to ensure accuracy.

3. Line 320, and 359 mention acceptability of DP but the methods of how this was assessed and the data to support this are missing.

*** Reviewer #4: 

Monthly sulfadoxine/pyrimethamine-amodiaquine or Dihydroartemisinin-piperaquine as malaria chemoprevention in young children with sickle cell anemia: a randomized controlled trial

Based on the rationale that there is paucity of high-quality comparative effectiveness studies of malaria chemoprevention regimens in African children with sickle cell anaemia (SCA), Taylor and colleagues report results of a trial to evaluate the efficacy of monthly sulfadoxine/pyrimethamine-amodiaquine (SPAQ), or monthly dihydroartemisinin-piperaquine (DP), with daily proguanil as the standard of care, for chemoprevention in children with SCA. The authors report comparable cumulative incidence of clinical malaria and cumulative incidence of painful events as primary and secondary outcome parameters, but slightly lower incidence of dactylitis and P. falciparum parasitaemia in the DP-assigned group.

The authors, in the introduction and rationale, rightly refer to the absence of data on the comparative effect of antimalarial chemoprophylactic regimens as well as lack of consensus on the optimal chemoprophylactic regimen in SCA, which is a population with poor malaria outcomes in endemic areas and in which chemoprevention is important. In this respect, the study by Taylor et. al., is potentially welcome and important contribution to the field. 

However, while preventive efficacy of chemoprophylactic agents, the main objective(s) of this study is justified, the safety of repeated antimalarial drug administration specifically in such a population is - I dare say - no less important. From this perspective, several opportunities of this study to demonstrate data that could fill in some of the pressing gaps in knowledge on the (critical) questions surrounding safety of repeated antimalarial drug administration in children in endemic areas in general, and in SCA children in particular, may have been lost.

1. Provide more details on the study site "routine SCA clinic" from where participants were recruited and in addition, include details on the routine SCA care practices (including use of folic acid etc) at the site.

2. Provide a rationale for performing certain investigations (e.g., repeat ECG) in only a selected subset of DP recipients (and why not across the three groups)

3. Per the manuscript (page 9) complete blood count was done every three months. It is surprising that authors present data for only hemoglobin, especially where other parameters (total WBC, platelets) are known to exhibit distinct characteristics in SCA and are linked to the pathophysiology of VOC/painful events as well as other significant SCA complications (e.g., stroke; transfusion requirements etc). Furthermore, the (historical) relationship between amodiaquine and neutropenia/agranulocytosis especially within the context of prophylaxis, is also well known. Given the above, I would strongly encourage authors to show the data for these other hematological parameters as this will strengthen the manuscript along both the safety and pathophysiology domains. In this respect, authors should present not only aggregated summaries but also a) changes from baseline; b) individual outlier data; as well as c) relationships (if any) between the data and the VOC/dactylitis outcome measure(s).

4. Authors should provide more complete information on participants in the "unanticipated crossover" group (number of participants, reasons for the intervention switch etc); suggest to (indeed required, per CONSORT Guidelines) to indicate such a 'flow' in the Trial Profile

5. There is a discrepancy in the number of children in whom followup ECG was conducted (see pages 9 vs 16).

6. The authors report more than 50% incidence of QTc prolongation in (an albeit limited) number of DP recipients. This is an important safety signal that should not, in my opinion, be ignored especially given the known relationship between DP and QTc interval prolongation. Importantly, the statement "given the observed safety of DP….."in the discussion (page 17) is not factually accurate.

7. It is premature to conclude on the safety of DP for prophylaxis in SCA children given the findings and the discussion should be substantially revised to take this into consideration.

8. The summary statement that "our results provide a rationale for the wider use of DP with appropriate monitoring as a routine component of care for children with SCA in malaria endemic settings" is not supported by the data.

9. Other comment: kindly use appropriate classification nomenclature for dactylitis (?hematologic event)

***

[LINK]

---

## [Decision Letter · Decision Letter 2]

14 Jun 2022

Dear Dr. Taylor,

Thank you very much for submitting your revised manuscript "Monthly sulfadoxine/pyrimethamine-amodiaquine or dihydroartemisinin-piperaquine as malaria chemoprevention in young Kenyan children with sickle cell anemia: A randomized controlled trial" (PMEDICINE-D-22-00862R2) for consideration at PLOS Medicine. 

Your paper was discussed with our academic editor and re-seen by three independent reviewers, including our statistical reviewer. The reviews are appended at the bottom of this email and any accompanying reviewer attachments can be seen via the link below:

[LINK]

In the light of these reviews, we will again be unable to accept the manuscript for publication in the journal in its current form, but we would like to invite you to submit a further revised version that addresses the reviewers' and editors' comments fully. You will recognize that we cannot make a decision on possible publication until we have seen the revised manuscript and your response, and we may seek re-review by one or more of the reviewers. 

We hope to receive your revised manuscript by Jul 08 2022 11:59PM. Please email us (plosmedicine@plos.org) if you have any questions or concerns.

Please let me know if you have any questions, and we look forward to receiving your revised manuscript. 

Sincerely,

Richard Turner, PhD

Senior editor, PLOS Medicine

rturner@plos.org

As mentioned previously, please adapt the data statement (submission form) so as to comply with PLOS' data policy, https://journals.plos.org/plosmedicine/s/data-availability.

Noting the comments from one referee, please report additional measures as appropriate to their inclusion in the study protocol/SAP (i.e., identifying comparisons as post-hoc if needed). 

Please add a few words to the abstract to identify the trial site. 

Please correct the referencing at lines 383-4.

Throughout the text, please move reference call-outs before punctuation (e.g., "... occlusive complications [6,7]."; noting the absence of spaces within the square brackets). 

Noting reference 29 and others, please use the journal name abbreviation "PLoS ONE".

Please correct the spelling of the first author's name for reference 36.

We did not find a completed CONSORT checklist with your revision and ask that you include this with your next revision. 

In the checklist, please refer to individual items by section (e.g., 'Methods') and paragraph number, not by line or page numbers as these generally change in the event of publication. 

Comments from academic editor: 

It may be most efficient to ask the authors to provide an additional direct comparison of

a) Text: Incidence of Pf infections in the DP vs SP-AQ arm

b) Text: Incidence of dactylitis in the DP vs SP-AQ arm

c) Table 1 and 2: add direct comparisons of DP vs SP-AQ arm

That should address “comparisons throughout the paper”.

The authors should review their conclusions (abstract, summary, and discussion) if warranted by the findings.

Comments from the reviewers:

*** Reviewer #1: 

Thank you to the authors for addressing my previous comments well. I would still recommend briefly noting the deviation from which endpoints were specified in the SAP compared to ct.gov (perhaps using more diplomatic language than in the response to comment 28!) but will leave that as a suggestion to the editor. Other that that I have no further issues to raise.

*** Reviewer #2: 

I think the presentation of results is misleading in the sense that the two interventions are being compared based on 

comparing the P-values for the difference of each with control, for example: "The incidence of P. falciparum infection relative to daily

Proguanil was similar in the monthly SP-AQ group (IRR 0.46; 95% CI: 0.17—1.20; p=0.13) but reduced with monthly DP (IRR 0.21; 9 5% CI: 0.08—0.56; p=0.002)." But it seems unlikely that the efficacy differs between the two interventions. Similarly, the rate ratio comparing the incidence of dactylis (blocked blood flow in the hands and feet causing painful swelling) in the DP group with the proguanil group was 0.47 (95%CI 0.23 - 0.96), and for the SPAQ group compared to proguanil, 0.78 (95%CI 0.41 - 1.50), it seems unlikely there could be a statistically significant difference between these effects. The difference in mortality is also likely to be a chance finding. There needs to be a head-on comparison of the two groups. This applies to comparisons throughout the paper. 

*** Reviewer #3: 

Two minor comments: 

1. The primary outcome could also lead the concluding parts of the abstract and the discussion. The authors can first highlight lack of difference in the primary outcome, probably mention the reason why, and then go on to mention important significant secondary outcomes like parasite prevalence, like in the statement below.

"Although there was no significant difference in the incidence of clinical malaria between the three treatment arms, possibly due to reduced malaria transmission intensity, DP was associated with……….then go on to mention important secondary outcomes that were significant."

2. Line 371, I suggest that "acceptability" be removed from the statement. To me, data presented in supplemental table 1 supports adherence not acceptability.

***

[LINK]

---

## [Decision Letter · Decision Letter 3]

10 Aug 2022

Dear Dr. Taylor,

Thank you very much for re-submitting your manuscript "Monthly sulfadoxine/pyrimethamine-amodiaquine or dihydroartemisinin-piperaquine as malaria chemoprevention in young Kenyan children with sickle cell anemia: A randomized controlled trial" (PMEDICINE-D-22-00862R3) for review by PLOS Medicine.

I have discussed the paper with my colleagues and the academic editor and it was also seen again by one reviewer. I am pleased to say that provided the remaining editorial and production issues are dealt with we are planning to accept the paper for publication in the journal.

[LINK]

We look forward to receiving the revised manuscript by Aug 17 2022 11:59PM.   

Sincerely,

Caitlin Moyer, PhD

Associate Editor 

PLOS Medicine

plosmedicine.org

Requests from Editors:

1. Data availability statement: The Data Availability Statement (DAS) requires revision. Please spell out “ICF” in the statement rather than using an acronym. It would be helpful to include a general web address where interested researchers can learn more about the dataset/ accessing the data. Please note that a study author cannot be the contact person for the data.

2. Funding statement: Please add the following information to the funding statement: “SP-AQ was supplied free of charge to the trial by Guilin Pharmaceuticals, which had no role in the design, conduct, analysis, reporting, or decision to report the results.”

3. Throughout: Please be sure that all abbreviations and acronyms are defined in the text at their first point of use (both for the Abstract and the main text).

4. Abstract: Background: We suggest an additional sentence providing context on the current chemoprevention practices or similar.

5. Abstract: Line 40: Please define HbSS at first use in the text. Please describe some brief information on the inclusion criteria/recruitment of the children with HbSS. Please note in the abstract that children were followed up monthly for 12 months.

6. Abstract: Line 45: Please provide a brief description of painful events and how this outcome was determined. Please list other secondary outcomes, including: parasitologic outcomes and hematologic outcomes.

7. Abstract: Line 58-59: Please point out that the differences in numbers of deaths between groups did not reach statistical significance.

8. Abstract: Conclusions: We suggest revising the first sentence to address the study implications without overreaching what can be concluded from the data, commenting on the primary outcome first and then key secondary findings; the phrase "In this study, we observed ..." may be useful. We suggest: In this study, we observed that monthly malaria chemoprevention with monthly SP-AQ or monthly DP did not significantly reduce incidence of clinical malaria compared with daily Proguanil. Monthly DP was associated with a lower rate of dactylitis and infection with P. falciparum relative to the Proguanil group, despite limited malaria transmission during the study period.” Please note that there is usually a distinction in the language in terms of causal vs. associational for primary and secondary trial outcomes. It would be beneficial to use associational language throughout when describing secondary outcomes.

9. Author summary: Line 93-94: Please rephrase slightly to: “...a higher rate of deaths occured in the SP-AQ group, but this did not reach statistical significance.” or similar.

10. Author summary: Line 98-99: Please revise to: “The observed greater number of deaths among children with SCA in the SP-AQ treated group was unexpected, and while this association did not reach statistical significance, further evaluation is warranted.” or similar.

11. Methods: Line 145-146: Please revise to: “"This study is reported as per the Consolidated Standards of Reporting Trials (CONSORT) guideline (S1 Checklist)."

12. Methods: Please provide additional details on eligibility in the main text of the Methods.

13. Methods: Lines 196 and 209: Please make sure that a complete listing of all primary and secondary outcomes, and safety outcomes, is included here.

14. Methods: Line 223-229: Please provide an explanation for the cross-over from SP-AQ to Proguanil.

15. Methods: Line 242-244: Please describe changes to the protocol, e.g. those prompted by the pandemic, in the text. Please mention and explain the reasons for the modifications including: exclusion of children with low hemoglobin, the collection of baseline laboratory testing at screening rather than enrollment, and the enhanced monthly hemoglobin monitoring, SAE review, and patient referral. Please complete and include the CONSERVE checklist, if relevant (https://www.equator-network.org/reporting-guidelines/guidelines-for-reporting-trial-protocols-and-completed-trials-modified-due-to-the-covid-19-pandemic-and-other-extenuating-circumstances-the-conserve-2021-statement/)

16. Results: Line 302: Please revise to: “...DP was associated with reduced rates of asymptomatic parasitization…”

17. Results: Line 315: Please revise to: “...did not modify the association between DP and dactylitis…”

18. Results: Please report p values in the text throughout, where appropriate, for example at lines 318-333.

19. Results: Lines 329-333: Please revise to indicate that the observed increase in risk of death among SP-AQ recipients (compared with the Proguanil or with the DP treated groups) was not statistically significant.

20. Discussion: Line 357-358: Please revise to: “...monthly DP was associated with significantly lower incidence of asymptomatic parasitization.”

21. Discussion: Line 365: Please revise to: “Monthly DP was associated with a reduced rate of episodes of dactylitis compared to daily Proguanil…”

22. Discussion: Line 398-399: It is not clear that the data support this statement, and we suggest removing it.

23. Discussion: line 426-427: Please mention the primary study outcome up front in the Conclusion paragraph. Please revise the sentence to: “...monthly DP was observed to be acceptable and was associated with reduced incidence of dactylitis and asymptomatic P. falciparum infection…”

24. Discussion: Line 429: We suggest removing: “and was associated with unacceptable morbidity.”

25. Figure 1: Please define all abbreviations used in the legend. 

26. Figure 2: Please define all abbreviations used in the legend. On the axis, please make it clear that the point estimates are relative for each SP-AQ or DP relative to Proguanil, not relative to each other. 

27. Table 1: It seems as though p values should not be needed as participants were randomized.

28. Table 3: Where reporting the p value for asymptomatic P. falciparum infection, please do not report as p<0.01. Please report the exact p value, unless p<0.001.

29. Reference 29, 34, and 36: Please change the journal to PLoS One.

30. CONSORT checklist: Thank you for including the CONSORT checklist. On the checklist, please make it clear you are referring to paragraph numbers within sections. For “Protocol” please report the location as Supporting Information files, or similar. For “Funding” please report the location as Funding statement, or similar.

31. Study Protocol: Thank you for including a copy of the study protocol. Please confirm that the image used in Appendix A-E is not reproduced from other sources that are not CC-BY. Please see https://journals.plos.org/plosmedicine/s/figures#loc-licenses-and-copyright for more information.

32. Supplemental Methods: Please move the description of inclusion/exclusion criteria and sample size considerations to the main text.

33. Supplemental Figure: Please define all abbreviations used in a descriptive legend.

34. Supporting information Table S6 and S7: Please define “-” and “(-,-)” in the legend.

35. Supporting information: Please check reference list formatting. Please use the "Vancouver" style for reference formatting, and see our website for other reference guidelines https://journals.plos.org/plosmedicine/s/submission-guidelines#loc-references

Comments from Reviewers:

Reviewer #5: This is a well written paper. 

The authors have adequately addressed the comments made by the reviewers.

I would suggest adding a column for p-values in Table 1.

[LINK]

---

## [Editor Report · Decision Letter 4]

26 Aug 2022

Dear Dr Taylor, 

On behalf of my colleagues and the Academic Editor, Lorenz von Seidlein, I am pleased to inform you that we have agreed to publish your manuscript "Monthly sulfadoxine/pyrimethamine-amodiaquine or dihydroartemisinin-piperaquine as malaria chemoprevention in young Kenyan children with sickle cell anemia: A randomized controlled trial" (PMEDICINE-D-22-00862R4) in PLOS Medicine.

Please also address the following editorial requests:

-Abstract: Lines 60-63: Please revise to: “Serious adverse events were common and distributed between groups, though compared to daily Proguanil (n=2), more children died receiving monthly SP-AQ (n=7; Hazard Ratio [HR] 5.44; 95% CI: 0.92—32.11) but not DP (n=1; HR 0.61; 95% CI 0.04—9.22), though differences did not reach statistical significance for either SP-AQ or DP.” Please also add p values for these results.

-Results: Line 356-360: Please revise to: “Ten children died: 2 receiving daily Proguanil, 7 receiving SP-AQ, and 1 receiving DP (Table 4); relative to Proguanil (cumulative IR [CIR] 2.2%), the risk of death was elevated in SP-AQ recipients, though this did not reach statistical significance (CIR 11.3%; Hazard Ratio [HR] 5.44; 95% CI 0.92 – 32.11; p=0.064), and was not elevated in DP recipients (CIR 1.3%; HR 0.61; 95% CI 0.04 – 9.22; p=0.89).

Discussion: Lines 385-387: Please revise to: “Compared to Proguanil, neither monthly SP-AQ nor DP reduced the incidence of the primary outcome clinical malaria, though monthly DP did significantly reduce the incidence of asymptomatic parasitization.”

-Discussion: Lines 458-460: Please revise to: “Monthly SP-AQ had no measurable benefit compared to daily Proguanil, and we observed a greater number of deaths in this group, though mortality rate differences were not statistically significant.”

-Figure 1: In the legend, please define QTcF, Hb, ALT, LAR.

-Study Protocol: Please remove or replace any image (e.g. in the Study Protocol) for which the source and license cannot be determined. Please see https://journals.plos.org/plosmedicine/s/licenses-and-copyright for further information.

-References: 

--Reference 2: Please abbreviate as Lancet Glob Health

--Reference 19: Please abbreviate as Lancet Haematol

--Reference 6 and 25: Please abbreviate as Br Med J

PRESS

Sincerely, 

Caitlin Moyer, Ph.D. 

Associate Editor 

PLOS Medicine